# Inappropriate Prescribing of Antibiotics to Pediatric Patients Receiving Medicaid: Comparison of High-Volume and Non-High-Volume Antibiotic Prescribers—Kentucky, 2019

**DOI:** 10.3390/healthcare11162307

**Published:** 2023-08-16

**Authors:** Bethany A. Wattles, Michael J. Smith, Yana Feygin, Kahir Jawad, Andrea Flinchum, Brittany Corley, Kevin B. Spicer

**Affiliations:** 1Department of Pediatrics, University of Louisville School of Medicine, Louisville, KY 40202, USA; 2Department of Pediatrics, Duke University Medical Center and Duke Center for Antimicrobial Stewardship and Infection Prevention, Durham, NC 27710, USA; 3Department of Pediatrics, Norton Children’s and University of Louisville School of Medicine, Louisville, KY 40202, USA; 4Healthcare-Associated Infection/Antibiotic Resistance Prevention Program, Division of Epidemiology and Health Planning, Kentucky Department for Public Health, Frankfort, KY 40621, USA; 5Division of Healthcare Quality Promotion, National Center for Emerging and Zoonotic Infectious Diseases, Centers for Disease Control and Prevention (CDC), Atlanta, GA 30333, USA

**Keywords:** antibiotic prescribing, Medicaid, antibiotic stewardship, outpatient

## Abstract

Inappropriate antibiotic prescribing to pediatric Medicaid patients was compared among high-volume and non-high-volume prescribers. High-volume prescribers had a higher percentage of inappropriate prescriptions than non-high-volume prescribers (17.2% versus 15.8%, *p* = 0.005). Targeting high-volume prescribers for stewardship efforts is a practical approach to reducing outpatient antibiotic prescribing that also captures inappropriate use.

## 1. Introduction 

Antimicrobial resistance (AMR) is a growing public health threat, and a key driver of resistance is antibiotic use [1]. The World Health Organization (WHO) has declared AMR as “one of the top 10 global public health threats facing humanity,” and identified the overuse and misuse of antimicrobials as “the main drivers in the development of drug-resistant pathogens” [2]. According to the Centers for Disease Control and Prevention (CDC), more than 2.9 million antibiotic-resistant infections occur in the United States (US) each year, and more than 35,000 people die as a result [1]. The majority of antibiotic prescribing occurs in the outpatient setting and an estimated 30% of outpatient prescriptions are unnecessary [1,3]. There are specific concerns related to antibiotic prescribing for Medicaid recipients. For example, a review of prescribing among the Medicaid population in New York found that overall the potential rate of inappropriate prescribing for acute respiratory infections was greater than 50% [4]. Additionally, a study reviewing national Medicaid claims from 2004 through 2013 found that 28% of nearly 300 million antibiotic prescriptions were not associated with a recent visit and an additional 17% were not associated with a visit that included a diagnosis of an infectious illness [5]. National agencies, public health departments, healthcare organizations, and payors have implemented antibiotic stewardship interventions to improve antibiotic use and mitigate the threat of antibiotic resistance. Some stewardship efforts, including U.S. annual reports on Outpatient Antibiotic Prescriptions by the CDC, use community pharmacy dispensing data to assess the volume of antibiotic prescriptions filled across the country [6,7]. Providing education to high-prescribing providers has been noted to reduce total and prolonged-duration antibiotic prescription and drug costs [8]. However, volume-based data do not allow for assessment of the appropriateness of antibiotic prescribing. Alternatively, healthcare claims data have been used extensively to quantify the appropriateness of antibiotic prescribing [9,10], using antibiotic pharmacy claims and clinical diagnoses from associated medical claims. These studies of the appropriateness of antibiotic prescribing, however, do not typically include information on overall antibiotic prescribing behaviors of individual providers, which may be important in developing more general interventions to address prescribing.

Kentucky has consistently had one of the highest rates of outpatient antibiotic prescribing in the nation since annual statewide reporting became publicly available from the CDC in 2011 [6,7]. Prior to 2020, Kentucky consistently ranked 2nd only to West Virginia in the rate of antibiotic prescribing in the outpatient setting, except for one year when Kentucky ranked 1st [6]. We sought to compare top antibiotic prescribers, identified by volume-based data, to other providers by rates of inappropriate prescribing using pediatric Medicaid claims data for this high-prescribing state. 

## 2. Materials and Methods

The CDC provided a subset of state public health departments, including the Kentucky Department for Public Health (KDPH), with a list of the top 10% of antibiotic prescribers, by specialty type, identified using 2019 IQVIA Xponent data. IQVIA is a commercial entity with a focus on data analytics, technology, and clinical research in the healthcare sector [11]. The Xponent data include prescriber-level information on dispensed drug prescriptions (i.e., “volume-based”) from a number of pharmacy types and covers up to 93% of the retail market [11]. The IQVIA Xponent top 10% of antibiotic prescribers were categorized as “high-volume prescribers” in analyses. To obtain additional information about antibiotic prescribing, we used Kentucky Medicaid claims from 2019 for children under 20 years of age to identify antibiotic prescriptions and associated medical claims. Medicaid claims were linked to medical visits within the three days prior (a method commonly used in claims-based research [9,10]) and diagnoses were categorized using a previously validated [9,10], mutually exclusive, tiered ICD-10 classification scheme in which a diagnosis is classified as “always” appropriate for an antibiotic (e.g., bacterial pneumonia, streptococcal pharyngitis), “sometimes” appropriate for an antibiotic (e.g., acute sinusitis, acute otitis media), or “inappropriate” for an antibiotic (e.g., acute upper respiratory infection, acute bronchitis). Prescriptions were classified as “not associated with an indication” if there was no medical claim identified within the three days prior; these prescriptions were not labeled as “inappropriate”. Further details of the classification scheme used, and its application to Medicaid claims data, are available in our previous publication [9]. Medicaid claims data were aggregated at the prescriber-level, using National Provider Identification (NPI) numbers to determine individual providers’ rates of inappropriate antibiotic prescribing. Prescribers were included if they wrote at least 12 antibiotic prescriptions to children insured by Kentucky Medicaid in 2019. 

High-volume antibiotic prescribers identified in the IQVIA Xponent data were matched to the Medicaid data using NPI numbers. “High-volume prescribers” were defined as those included in the top 10% list from IQVIA data. Any prescribers included from the Kentucky Medicaid data that were not included in the “High-volume prescribers” category (identified by the IQVIA top 10% data), were categorized as “non-high-volume prescribers”. The provider type was obtained from Kentucky Medicaid claims and categorized prescribers into one of the following groups: general practitioner, nurse practitioner, physician assistant, pediatrician, and other. Dentists were excluded from both prescriber lists, as associated dental diagnoses are not reliably available in our Medicaid data files. Descriptive statistics were used to summarize the cohort, with frequency and percentages used for categorical data, and median and interquartile range (IQR) utilized for continuous variables. Non-parametric Mann–Whitney U tests were used to compare the distributions (or median percentages) of inappropriate antibiotic prescriptions between prescriber groups. 

## 3. Results

There were 1083 providers identified as high-volume prescribers in the 2019 IQVIA Xponent data, 878 (81.1%) of whom were included on the Medicaid prescriber list. There were 5149 prescribers on the Medicaid list, of which 878 (17.1%) were on the IQVIA list (of high-volume antibiotic prescribers). Overall, high-volume prescribers provided a median of 245 (IQR 107-428) antibiotic prescriptions to children insured by Kentucky Medicaid in 2019, compared to non-high-volume prescribers who provided a median of 46 antibiotic prescriptions (IQR 22-106). A total of 676,476 Medicaid antibiotic prescriptions were included, 133,662 (19.8%) of which were determined to be inappropriate. High-volume prescribers wrote 293,330 (43.4%) of all antibiotic prescriptions included. Among antibiotic prescriptions written by high-volume prescribers, 62,557 (21.3%) were inappropriate, representing 46.8% of all inappropriate prescriptions.

Considering all prescribers, across provider groups, high-volume antibiotic prescribers had a higher percentage of inappropriate prescriptions to pediatric Medicaid patients than did non-high-volume prescribers (17.2% versus 15.8%, *p* = 0.005) (Table 1). There was no statistically significant difference in the median of the percentage of inappropriate antibiotic prescriptions between high-volume and non-high-volume antibiotic prescribers among the following groups of providers: nurse practitioners, physician assistants, and pediatricians (Table 2). The percentage of inappropriate antibiotic prescriptions was higher for high-volume antibiotic prescribers than for non-high-volume prescribers among those categorized as general practitioners (21.1% versus 15.8%, *p* < 0.001). There was also a trend toward higher inappropriate antibiotic prescribing for high-volume antibiotic prescribers in the “other” category (23.5% versus 19.1%, *p* = 0.07).

## 4. Discussion

High-volume antibiotic prescribers, who made up 17% of all prescribers evaluated, were responsible for nearly half (47%) of all inappropriate antibiotic prescriptions to children insured by Kentucky Medicaid. Although rates of inappropriate prescribing by high-volume and non-high-volume antibiotic prescribers were similar for most provider types, targeting the top 10% of prescribers by volume for stewardship efforts has the potential to be a high-impact approach to reducing unnecessary outpatient antibiotic prescribing given the high proportion of overall antibiotic prescriptions written by this group. This approach may be particularly useful if complementary data are available to allow an emphasis on high-volume prescribers with a higher likelihood of inappropriate prescribing [8]. 

In order to reduce unnecessary antibiotic prescribing and the subsequent development of antimicrobial resistance, much work in outpatient antibiotic stewardship has focused on reducing inappropriate antibiotic prescribing [3,9,10,12,13,14]. However, assigning an antibiotic prescription as “inappropriate” requires an associated medical diagnosis, which is unavailable in strictly volume-based prescribing or dispensing data sources. Our findings may be useful to public health officials and clinicians seeking to design methods to deliver outpatient antibiotic prescribing feedback to clinicians across a broad area (e.g., statewide or nationally) when limited to the availability of volume-based data or by data analysis support. Such feedback could be provided at an aggregate level, allowing for targeted education and other interventions that would require less intensive analysis than that necessary to provide feedback to individual providers on their specific performance relative to peers.

Our study found that high-volume prescribers had significantly higher rates of inappropriate prescribing, particularly among general practitioners. Among children insured by Kentucky Medicaid, general practitioners prescribe approximately 30–40% of all antibiotics, compared to approximately 10% of prescriptions by pediatricians [15], highlighting the importance of including general practitioners in pediatric antibiotic stewardship efforts. Also of note, while high-volume and non-high-volume prescribing nurse practitioners had similar rates of inappropriate prescriptions, nurse practitioners prescribed over 50% of all included antibiotic prescriptions. Consequently, this group of providers should be included in future antibiotic stewardship efforts in Kentucky since even a small reduction in inappropriate prescribing in this group could have substantial impact on overall antibiotic prescribing. Additionally, future directions of this work would benefit from expansion to include patients of all ages.

Unnecessary and inappropriate antibiotic prescribing can directly impact individuals and can influence levels of resistance noted in the population. For example, a study evaluating the initial diagnosis of a urinary tract infection in children found that children with recent exposure to amoxicillin were more likely to have a urinary pathogen with ampicillin and amoxicillin-clavulanate resistance [16]. Similarly, a more recent study from Vietnam found that antibiotic use, often inappropriate, for acute respiratory infections was associated with noted resistance to cephalosporins, aminoglycosides, and fluoroquinolones among intestinal *Enterobacteriaceae* [17]. From the population perspective, a study of outpatient antibiotic prescribing from 2011 through 2014 in the United States found that antibiotic resistance appeared to be more related to low-intensity, broadly distributed use than to repeated, high-intensity use [18]. Such findings support complementary approaches that focus on improving appropriateness for individual providers and patients and on decreasing overall use by targeting high-prescribing providers. An approach that targets high-prescribing providers would potentially reduce initial antibiotic starts and should also impact appropriateness given the noted association of volume with inappropriateness in the current study.

A similar study in adults found that the top 10% of antibiotic prescribers wrote 24.4 million (41%) of the 59.4 million antibiotic prescriptions for Medicare Part D beneficiaries in the US in 2019 [19]. Our findings add to this work by assessing antibiotic prescribing to children and including inappropriateness of prescribing. Previous studies have utilized Medicaid claims data to identify variations in antibiotic prescribing [5,9]. Benefits of Medicaid data include a large, statewide population sample and ability to assess appropriateness of prescribing. However, limitations of Medicaid data include access to and timeliness of data, complexity of analysis, and a sample population not fully representative of patients across the state. Other limitations inherent to claims data include reliance on accurate billing data for identification of diagnoses and uncertainty of associated visits [9,19]. These known limitations of claims data highlight the public health importance of the current study; we have shown that less complicated and more easily accessible volume-based prescribing data can accurately identify providers who inappropriately prescribe antibiotics. Other limitations specific to this study include (1) use of only pediatric data for the Medicaid prescriber list, while the IQVIA data included top antibiotic prescribers to all ages; and (2) in both data sources, we were limited by existing classifications of prescriber types (e.g., general practitioner, nurse practitioner). 

In 2021, the KDPH, along with other states, utilized the top 10% IQVIA provider lists to send informational letters to top antibiotic prescribers. The current study, a collaboration among stewardship researchers and KDPH, suggests volume-based antibiotic use reporting is a practical approach to target state outpatient antibiotic stewardship efforts as it captures a substantial proportion of inappropriate prescribing. Additionally, including general practitioners and nurse practitioners in pediatric stewardship efforts has potential for high-impact reductions in unnecessary antibiotic prescribing. Efforts to reduce excess antibiotic prescribing are necessary and important in addressing the national and global threat of antimicrobial resistance. Further, implementing antibiotic stewardship efforts, such as prescriber feedback, on a broader scale (statewide, regional, or national) has greater potential to impact the overall threat of resistance. 

## 5. Conclusions

Medicaid claims and prescription dispensing data are useful for broad-scale outpatient antibiotic stewardship efforts. Targeting high-volume prescribers is a practical approach to provider feedback and also captures inappropriate prescribing. 

## Figures and Tables

**Table 1 healthcare-11-02307-t001:** Total and inappropriate antibiotic prescribing to pediatric patients receiving Medicaid, by provider type—Kentucky, 2019 ^a^.

	High-Volume Prescribers ^b^			Non-High-Volume Prescribers ^c^		
	N (%)	Total Antibiotic Prescribing, N (%)	Inappropriate Antibiotic Prescribing, N (%) ^d^	N (%)	Total Antibiotic Prescribing, N (%)	Inappropriate Antibiotic Prescribing, N (%) ^d^
Overall	878	293,330	62,557	4271	383,146	71,105
General Practitioner	237 (27.0)	76,091	19,932	1436 (33.6)	112,407	21,720
		(25.9)	(31.9)		(29.3)	(30.5)
Nurse Practitioner	468 (53.3)	162,332	30,585	1682 (39.4)	170,736	29,850
		(55.3)	(48.9)		(44.6)	(42.0)
Physician Assistant	81	24,940	5101	389	35,282	6620
	(9.2)	(8.5)	(8.2)	(9.1)	(9.2)	(9.3)
Pediatrician	41	20,286	4353	318	45,090	8606
	(4.7)	(6.9)	(7.0)	(7.4)	(11.8)	(12.1)
Other	51	9681	2586 (4.1)	446 (10.4)	19,631	4309
	(5.8)	(3.3)			(5.1)	(6.1)

^a^ Medicaid providers were included if they wrote at least 12 antibiotic prescriptions to children insured by Kentucky Medicaid in 2019. ^b^ High-Volume Prescribers = on top 10% list identified using IQVIA Xponent and included in Medicaid list; prescribing to pediatric Medicaid patients. ^c^ Non-High-Volume Prescribers = included in Medicaid list but not IQVIA top 10% list; prescribing to pediatric Medicaid patients. ^d^ Number of antibiotic prescriptions deemed inappropriate; percentage reflects percent of overall inappropriate prescriptions written by the provider type.

**Table 2 healthcare-11-02307-t002:** Median inappropriate antibiotic percentage and interquartile range, by provider type—Kentucky, 2019 ^a^.

	High-Volume Prescribers ^b^		Non-High-Volume Prescribers ^c^		
	N (%)	Inappropriate Antibiotic Percentage %, Median (IQR) ^d^	N (%)	Inappropriate Antibiotic Percentage %, Median (IQR) ^d^	Inappropriate Antibiotic Percentage, *p*-Value ^e^
Overall	878	17.2	4271	15.8	0.005
		(9.5, 29.1)		(8.3, 27.0)	
General Practitioner	237 (27.0)	21.1	1436 (33.6)	15.8	<0.001
		(11.9, 33.9)		(8.7, 26.7)	
Nurse Practitioner	468 (53.3)	15.4	1682 (39.4)	15.0	0.83
		(7.7, 25.7)		(7.7, 26.5)	
Physician Assistant	81 (9.2)	16.2	389 (9.1)	16.7	0.99
		(9.1, 25.6)		(9.5, 25.0)	
Pediatrician	41 (4.7)	17.2	318 (7.4)	14.9	0.39
		(10.1, 27.2)		(8.1, 26.6)	
Other	51 (5.8)	23.5	446 (10.4)	19.1	0.07
		(14.5, 33.2)		(8.8, 31.5)	

^a^ Medicaid providers were included if they wrote at least 12 antibiotic prescriptions to children insured by Kentucky Medicaid in 2019. ^b^ High-Volume Prescribers = on top 10% list identified using IQVIA Xponent and included in Medicaid list; prescribing to pediatric Medicaid patients. ^c^ Non-High-Volume Prescribers = included in Medicaid list but not IQVIA top 10% list; prescribing to pediatric Medicaid patients. ^d^ Median percentage (and IQR) of inappropriate prescriptions written by providers within each provider type. ^e^ *p*-value of Mann–Whitney U tests comparing inappropriate antibiotic median percentage between high-volume and other Medicaid prescribers. Bold was to signify statistically significant *p*-values (<0.05)

## Data Availability

Restrictions apply to the availability of the data. IQVIA data were obtained from the Kentucky Department for Public Health HAI/AR Prevention Program through a limited agreement with the Centers for Disease Control and Prevention. Medicaid data were obtained through a limited data use agreement.

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
