# Peer review of "Inappropriate Prescribing of Antibiotics to Pediatric Patients Receiving Medicaid: Comparison of High-Volume and Non-High-Volume Antibiotic Prescribers—Kentucky, 2019"

_healthcare, 2023, doi:10.3390/healthcare11162307_

Round 1

Reviewer 1 Report

Strengths of the paper:

Relevant and important topic: The paper addresses the important issue of inappropriate antibiotic prescribing to pediatric patients receiving Medicaid, which is a significant public health concern due to its association with antimicrobial resistance.

Comparison of high-volume and non-high-volume prescribers: The paper compares the rates of inappropriate prescribing between high-volume and non-high-volume prescribers, providing valuable insights into the potential impact of targeting high-volume prescribers for stewardship efforts.

Use of healthcare claims data: The paper utilizes Medicaid claims data to study antibiotic prescribing, which allows for a large, statewide population sample and the assessment of appropriateness of prescribing. This strengthens the study's validity and generalizability.

Clear methodology: The Materials & Methods section provides a clear description of the study's methodology, including the data sources, categorization of appropriateness, and statistical analyses used. This transparency enhances the replicability of the study.

Weaknesses of the paper:

Lack of detailed discussion on appropriateness criteria: Although the paper mentions the use of a tiered ICD-10 classification scheme to categorize appropriateness, it does not provide specific details on the criteria used or refer to the relevant study where the scheme was described. Providing more information on the appropriateness criteria would improve the transparency of the assessment process.

Incomplete referencing: The paper lacks proper referencing in some sections, such as the introduction where specific citations for CDC reports and the use of community pharmacy data are missing. Ensuring all references are properly cited enhances the credibility of the paper and allows readers to access the primary sources.

Limited discussion on limitations: While the Discussion section briefly mentions limitations inherent to claims data, it does not thoroughly discuss other potential limitations of the study. This could include factors such as data access and timeliness, representativeness of the Medicaid sample, or other potential biases. Addressing these limitations would strengthen the paper's transparency and help readers better understand the potential impact on the study's findings.

Insufficient contextualization of findings: The Discussion section could benefit from further contextualizing the findings within the broader literature on inappropriate antibiotic prescribing and stewardship efforts. This could involve discussing how the study's findings align with or differ from previous studies, providing insights into the implications for policy and practice, and identifying potential areas for future research.

Improvement suggestions:

Provide more detailed information on the appropriateness criteria used and refer to the relevant study where the classification scheme was described. This would enhance the transparency of the assessment process.

Ensure all references are properly cited throughout the paper, including specific citations for CDC reports and other relevant sources.

Discuss potential limitations of the study in more detail, addressing factors such as data access and timeliness, sample representativeness, and other potential biases. This would provide a more comprehensive assessment of the study's limitations.

Contextualize the findings within the existing literature on inappropriate antibiotic prescribing and stewardship efforts, discussing similarities, differences, and implications for policy and practice. This would provide a broader perspective and enhance the paper's contribution to the field.

Consider expanding the introduction to provide more specific statistics or examples to emphasize the magnitude of the problem of inappropriate antibiotic prescribing and its impact on antimicrobial resistance. This would help readers better understand the urgency and importance of the topic.

Style, grammar and punctuation can be improved upon proofreading. But not a major issue. 

Author Response

Improvement suggestions:

Provide more detailed information on the appropriateness criteria used and refer to the relevant study where the classification scheme was described. This would enhance the transparency of the assessment process.

Thank you, we have made additions to the methods section to address this feedback.

Ensure all references are properly cited throughout the paper, including specific citations for CDC reports and other relevant sources.

References 6 and 7 cite the CDC online reports and the initial publication using national pharmacy dispensing data. We have moved these references to the end of the sentence in the introduction to add clarity.

Discuss potential limitations of the study in more detail, addressing factors such as data access and timeliness, sample representativeness, and other potential biases. This would provide a more comprehensive assessment of the study's limitations.

Thank you for this suggestion, we have added the following to the limitations section: “Other limitations inherent to claims data include reliance on accurate billing data for identification of diagnoses, uncertainty of associated visit and prescribing data.9,21 These known limitations of claims data highlight the public health importance of the current study; we have shown that less complicated and more easily accessible volume-based prescribing data can accurately identify providers who inappropriately prescribe antibiotics. Other limitations specific to this study include 1) use of only pediatric data for the Medicaid prescriber list, while the IQVIA data included top antibiotic prescribers to all ages; and 2) in both data sources, we were limited by existing classifications of prescriber types (e.g. general practitioner, nurse practitioner).”

Contextualize the findings within the existing literature on inappropriate antibiotic prescribing and stewardship efforts, discussing similarities, differences, and implications for policy and practice. This would provide a broader perspective and enhance the paper's contribution to the field.

Thank you for this suggestion, we have significantly expanded the discussion and believe it now addresses these suggestions.

Consider expanding the introduction to provide more specific statistics or examples to emphasize the magnitude of the problem of inappropriate antibiotic prescribing and its impact on antimicrobial resistance. This would help readers better understand the urgency and importance of the topic.

Thank you, we have added the following to the introduction: “The World Health Organization (WHO) has declared AMR as “one of the top 10 global public health threats facing humanity,” and identified overuse and misuse of antimicrobials as “the main drivers in the development of drug-resistant pathogens.”2According to the Centers for Disease Control and Prevention (CDC), more than 2.9 million antibiotic-resistant infections occur in the United States (US) each year, and more than 35,000 people die as a result.1"

Reviewer 2 Report

Careful analysis of antibiotic prescriptions with a particular focus on prescribing appropriateness.

The topic finds its area of interest in the improvement of antibiotic appropriateness considering the emergence of the increase in antibiotic resistance at a global level. The paper is a careful analysis of antibiotic prescriptions with a particular focus on prescribing appropriateness. The study compared antibiotic prescribing in a group of high-volume prescribers and non-high-volume prescribers with the highest prescribing of antibiotics in the former group. Prescribing inappropriateness was greatest in high-volume prescribers.

The sample examined is large enough to validate the study. Antibiotic prescribing was studied using Medicaid indications.

The methodology of the study is clear and complete with statistical analysis.

The Authors refer to a categorization to define the prescriptive appropriateness of antibiotics, reporting a reference (5). It would be preferable to insert in "Materials and Methods" (page 1 line 40) the parameters and categories with which the Authors define "appropriate", "potentially appropriate" and "inappropriate" an antibiotic therapy, instead of just the bibliographic reference 5. About this, it would be appropriate to indicate other references to define the prescriptive appropriateness.

In the Materials and Methods Section it would be good to describe any prescribing differences in the groups examined (general practitioner, nurse practitioner, physician assistant, pediatrician, etc.) in order to better evaluate the data collected and correctly stratified in tables 1-2. These data could be part of further study in the paragraph of the Discussion.

It would be preferable to insert in "Materials and Methods" (page 1 line 40) the parameters and categories with which the Authors define "appropriate", "potentially appropriate" and "inappropriate" an antibiotic therapy, instead of just the bibliographic reference (5).

In the Discussion section, the results obtained regarding the antibiotic appropriateness according to the stratification in the different categories of prescribers should be analyzed in detail.

The results in the two groups examined, high-volume prescribers and non-high-volume prescribers, deserve more detailed analysis. The Authors should describe in the Materials and Methods Section the differences of the examined population as indicated in the legends of Tables 1-2.

It would be interesting to know which classes of antibiotics were most prescribed, considering that a pediatric and an adult population were examined. A detailed analysis of the results reported in the tables in the two groups of adult and pediatric population.

The Authors should better comment on the interesting statistic data about the prescribing inappropriateness of antibiotics found in the group of general practitioners.

At the end of the Discussion, reference is made to the possible effects that the prescribing appropriateness of antibiotics has on the control of the spread of antimicrobial resistance. Authors should stress this important concept and contextualize it in their work.

Author Response

Careful analysis of antibiotic prescriptions with a particular focus on prescribing appropriateness.

The topic finds its area of interest in the improvement of antibiotic appropriateness considering the emergence of the increase in antibiotic resistance at a global level. The paper is a careful analysis of antibiotic prescriptions with a particular focus on prescribing appropriateness. The study compared antibiotic prescribing in a group of high-volume prescribers and non-high-volume prescribers with the highest prescribing of antibiotics in the former group. Prescribing inappropriateness was greatest in high-volume prescribers. 

The sample examined is large enough to validate the study. Antibiotic prescribing was studied using Medicaid indications.

The methodology of the study is clear and complete with statistical analysis.

The Authors refer to a categorization to define the prescriptive appropriateness of antibiotics, reporting a reference (5). It would be preferable to insert in "Materials and Methods" (page 1 line 40) the parameters and categories with which the Authors define "appropriate", "potentially appropriate" and "inappropriate" an antibiotic therapy, instead of just the bibliographic reference 5. About this, it would be appropriate to indicate other references to define the prescriptive appropriateness.

Thank you, we have expanded the description of appropriateness categories and also cite the original study by Chua, et al. in 2019.

In the Materials and Methods section, it would be good to describe any prescribing differences in the groups examined (general practitioner, nurse practitioner, physician assistant, pediatrician, etc.) in order to better evaluate the data collected and correctly stratified in tables 1-2. These data could be part of further study in the paragraph of the Discussion.

Please see the following statement in the materials and methods section referring to provider type categories: “The provider type variable in Medicaid claims was used to categorize prescribers into one of the following groups: general practitioner, nurse practitioner, physician assistant, pediatrician, and other.”

And in the discussion: Additionally, in both data sources, we were limited by existing classifications of prescriber types (e.g. general practitioner, nurse practitioner).”

It would be preferable to insert in "Materials and Methods" (page 1 line 40) the parameters and categories with which the Authors define "appropriate", "potentially appropriate" and "inappropriate" an antibiotic therapy, instead of just the bibliographic reference (5).

Please see responses to reviewer 1.

In the Discussion section, the results obtained regarding the antibiotic appropriateness according to the stratification in the different categories of prescribers should be analyzed in detail.

Thank you for the suggestion, we have made the following addition to the discussion: Our study found that overall, and among general practitioners, high-volume prescribers had significantly higher rates of inappropriate prescribing. Among children insured by Kentucky Medicaid, general practitioners prescribe approximately 30-40% of all antibiotics, compared to only approximately 10% of prescriptions by pediatricians, highlighting the importance of including general practitioners in pediatric antibiotic stewardship efforts. Additionally, future directions of this work would benefit from expansion to include patients of all ages. Also of note, while High-Volume and Lower-Volume nurse practitioners had similar rates of inappropriate prescriptions, they prescribed over 50% of all included antibiotic prescriptions and should also be involved in future antibiotic stewardship efforts in Kentucky.”

The results in the two groups examined, high-volume prescribers and non-high-volume prescribers, deserve more detailed analysis. The Authors should describe in the Materials and Methods Section the differences of the examined population as indicated in the legends of Tables 1-2.

Thank you for this suggestion, we have added this information to the methods section.

It would be interesting to know which classes of antibiotics were most prescribed, considering that a pediatric and an adult population were examined. A detailed analysis of the results reported in the tables in the two groups of adult and pediatric population.

Unfortunately, analysis of specific medications was beyond the scope of this study.

The Authors should better comment on the interesting statistic data about the prescribing inappropriateness of antibiotics found in the group of general practitioners.

Thank you for this suggestion, please see the response above.

At the end of the Discussion, reference is made to the possible effects that the prescribing appropriateness of antibiotics has on the control of the spread of antimicrobial resistance. Authors should stress this important concept and contextualize it in their work.

We have added the following additional statement in response to this suggestion: “Further, implementing antibiotic stewardship efforts, such as prescriber feedback, on a broader scale (statewide, regional, or national) has greater potential to impact the overall threat of resistance.”

Round 2

Reviewer 1 Report

Authors have addressed most of the comments. Please go over one more round of editing for style and punctuation. 

Authors have addressed most of the comments. Please go over one more round of editing for style and punctuation.